# Aqueous Leaf Extracts of Peppermint (*Mentha* × *piperita*) and White Snakeroot (*Ageratina altissima*) Exhibit Antibacterial and Antiviral Activity

**DOI:** 10.3390/microorganisms14010080

**Published:** 2025-12-30

**Authors:** Mackenzie E. Yurchiak, Shea Bailey, Aarish H. Sakib, Macy M. Smith, Rachael Lally, Jacob W. DuBrava, Keely M. Roe, Orna Stuart, Abigail E. Shafier, Juhee Kim, Lauren D. Susick, Lia Prassas, Audrey L. Voss, Grace C. O’Malley, Sofia Calvo, Marek B. Magnus, Sean T. Berthrong, Anne M. Wilson, Michael P. Trombley, Ashlee H. Tietje, Christopher C. Stobart

**Affiliations:** 1Department of Biological Sciences, Butler University, Indianapolis, IN 46208, USAsberthro@butler.edu (S.T.B.); mtromble@butler.edu (M.P.T.); atietje@butler.edu (A.H.T.); 2Clowes Department of Chemistry and Biochemistry, Butler University, Indianapolis, IN 46208, USA; amwilson@butler.edu

**Keywords:** antiviral, cancer, phytochemistry, respiratory syncytial virus, antibacterial

## Abstract

With new emerging diseases such as COVID-19 and an increasing incidence of cancer, there remains a significant need for investigating new therapeutic options to treat a wide range of ailments and disorders. Peppermint (*Mentha* × *piperita*) and white snakeroot (*Ageratina altissima*) have been used medicinally by native people in the Midwestern United States for centuries. However, the antiproliferative and antimicrobial properties of the aqueous extracts of these plants remain unclear. In this study, we evaluate the therapeutic potential of peppermint and white snakeroot aqueous leaf extracts by examining their activity against mammalian cancer cells, bacteria, and viruses. Both peppermint and snakeroot extracts showed no reductions in viability at concentrations lower than 25 mg/mL and 10 mg/mL, respectively, in two different cancer lines, HEp-2 and DBT-9 cells, in vitro. While treatment with the snakeroot extract resulted in significant disruption to cytoskeletal organization in HEp-2 cells at a concentration of 10 mg/mL, peppermint and snakeroot extracts did not have a major impact on the viability or proliferation of the cancer cell lines tested. Peppermint and snakeroot were then evaluated for antibacterial activity against four different bacterial pathogens. Significant inhibition of bacterial replication was observed for *E. coli* (at concentrations greater than 0.1 mg/mL) and *S. aureus* (at concentrations greater than 1 mg/mL) treated with either peppermint or snakeroot extracts. No significant activity was observed against the bacterial strains *P. aeruginosa* and *S. pyogenes*. Peppermint (EC_50_ = 2.36 mg/mL) and snakeroot (EC_50_ = 2.64 mg/mL) significantly reduce infectivity and replication (at concentrations above 0.2 mg/mL) of the major human pathogen, human respiratory syncytial virus (hRSV). However, testing for antiviral activity against a mouse coronavirus (murine hepatitis virus, MHV) showed no impact on replication at concentrations up to 2.5 mg/mL. Lastly, chemical analysis of the extracts identified several prominent compounds, which were subsequently evaluated for their biological contributions to the observed plant extract phenotypes. Two of the identified compounds, 1,8-cineole (Eucalyptol) and menthol, show significant antimicrobial activity. We report that aqueous extracts of peppermint and white snakeroot exhibit specific antibacterial and antiviral activities that support further investigation for therapeutic potential.

## 1. Introduction

Despite significant recent advances in healthcare, there remains a significant need for the discovery of new therapeutics to treat a wide range of ailments, including cancer and infectious diseases. Advances in the early screening and detection of cancers have led to improved outcomes for nearly all types of cancer; however, the collective incidence of cancer has continued to increase, highlighting the importance of continuing to investigate novel therapies [1,2]. In addition to increases in the incidence of cancer, there have also been significant increases in both antibiotic resistance among extant human microbial pathogens, such as the proliferation of multi-drug-resistant tuberculosis strains, and in the emergence of new human infectious diseases, as evidenced by SARS-CoV-2, the causative agent of COVID-19 [3,4,5].

It is estimated that up to half of all therapeutics used today are derived from natural plant products [6]. However, only about 10% of all plants are believed to have been investigated for potential therapeutic properties [6,7]. Identifying plants that harbor phytochemicals with therapeutic potential may reveal new directions for drug development, addressing the ever-increasing threats of increased cancer incidence, antibiotic resistance, and both existing and emerging infectious diseases. Notably, there remains a significant need to identify potential antivirals to treat a wide array of common viral pathogens [8]. For example, human respiratory syncytial virus (hRSV) remains a major cause of hospitalizations and mortality among infants and the elderly, for which there are no current specific antiviral options available [9].

Peppermint (*Mentha* × *piperita*) and white snakeroot (*Ageratina altissima*) have been used traditionally by Native Americans to treat a variety of ailments and illnesses. Peppermint is an herbaceous and perennial herb of the family *Lamiaceae*, which is native to Europe, but can now be found worldwide [10]. Peppermint and peppermint oil have a long history of being used medicinally to treat a variety of respiratory and digestive ailments [11,12]. Chemical analysis of various leaf extract preparations from peppermint has shown that the major constituents of the plant include many terpenoids and phenolics, which are believed to be responsible for its biological activity [13]. In addition, peppermint is not widely described as harboring extensive anticancer activity; however, some limited studies have described the antiproliferative activity of several constituents (such as rosmarinic acid and menthol) [14,15]. While peppermint extracts have been extensively studied for both their antibacterial and antiviral properties, it remains unclear how effective aqueous extracts of peppermint are towards cancer cell proliferation, as well as the major human respiratory pathogen, the human respiratory syncytial virus (hRSV), and what role the chemical components of the plant have on any observed biologic or therapeutic activities [12,16,17].

White snakeroot is an herbaceous perennial of the family *Asteraceae* that is native to eastern North America but has now been identified as an invasive species in many other regions, including Europe and Asia [18,19,20,21]. The plant grows to heights of up to 1.5 m and is most notably identified by the formation of composite heads of small white flowers. Snakeroot is considered toxic to mammals due to containing a fat-soluble toxin called tremetol, which concentrates in milk and can lead to a severe and potentially fatal disease called “milk sickness” if consumed [22]. However, Native Americans have reportedly used snakeroot to treat poisonous snake bites (the source of its common name), as well as several other diseases, including fever and urinary infections [23]. White snakeroot has been far less studied for its therapeutic potential due in part to its known toxicity in mammals. However, other *Ageratina* species have been explored for their wound healing, antioxidant, and antiproliferative effects. A study looking at aqueous extracts of *Ageratina pichinchensis* (Mexican snakeroot) showed that the plant did not induce significant cytotoxicity in several cancer cell lines, but was able to promote cell proliferation in an MRC-5 cell line [24]. Another study showed limited cytotoxicity in 4T1 breast cancer cells using alcohol extracts of *Ageratina havanensis* (Havana snakeroot or white mistflower) at high concentrations [25]. Collectively, these studies show potential inconsistencies in the activity of *Ageratina* extracts on cancer cell proliferation, warranting further investigation. Chemical analysis of snakeroot reveals several prominent types of benzofuran ketones (such as tremetone found in tremetol) and chromene glycosides [26,27]. The antimicrobial properties of *A. altissima* extracts remain unknown.

In this study, we investigated the therapeutic potential of extracts prepared from two plants commonly found throughout the Midwestern United States—peppermint and white snakeroot. The primary objective of this study is to determine if aqueous extracts of these plants harbor anticancer, antibacterial, or antiviral activity. Consistent with the traditional preparation of teas and tonics, leaf-based extracts were prepared in aqueous solution using only deionized water for the experiments described herein. Plant extracts of peppermint and snakeroot, and subsequently their major chemical constituents identified in this study, were evaluated for activity against two different cancer cell lines (HEp-2 and DBT-9), four species of bacteria (*E. coli*, *P. aeruginosa*, *S. aureus*, and *S. pyogenes*), and two different viruses, hRSV and murine hepatitis virus (MHV), a mouse coronavirus. We show here that peppermint and white snakeroot extracts exhibit significant antibacterial and antiviral activity against specific bacteria and viruses. No major impacts on viability or cell morphology were observed in either cancer line in response to either plant extract treatment. Lastly, we identified several compounds present in peppermint and snakeroot and evaluated their phenotypic contributions to the antimicrobial activity of the parent extracts. These studies support further investigation of peppermint and white snakeroot for use in antimicrobial therapies.

## 2. Materials and Methods

### 2.1. Plant Leaf Harvesting and Extract Preparation

Peppermint (*Mentha* × *piperita*) and white snakeroot (*Ageratina altissima*) leaves were harvested in September 2023 from the Butler University campus in Indianapolis, IN (39°50′19.853″ N, 86°10′21.221″ W). Plant specimens of peppermint and snakeroot were positively identified, and dried specimens were properly preserved, deposited in the Friesner Herbarium at Butler University (Accession numbers 20240122 and 20240112-0114, respectively), and are available for examination upon request. To prepare leaf extracts, the plant leaves were separated from stems and dried using a mechanical convection oven at approximately 32 °C (90 °F) before being coarsely chopped and steeped in deionized water at 95 °C to produce final extract solutions with concentrations of 0.1 g/mL. Plant extracts were aliquoted and frozen at −80 °C. Prior to use, the extract solutions were thawed completely, resuspended, filtered, and sterilized using a 0.2-micron filter.

### 2.2. GC–MS Analysis of Extracts and Compound Preparation

To analyze the major constituents of the peppermint and snakeroot extracts, 1 mL of each extract was mixed with 10 mL of methanol and extracted with 500 mL of ethyl acetate in the presence of 200 mg of anhydrous NaCl. The organic phase was separated and analyzed using a ThermoFisher Scientific TRACE 1310 GC-ISQ LT gas chromatography–mass spectrometry (GC–MS) instrument run (Waltham, MA, USA) with Xcalibur (v4.5) following a modified procedure described earlier [28,29]. Data were processed using FreeStyle 1.8, and key compounds were identified using the NIST compound library accessed through Compound Discover.

The identified compounds were subsequently purchased for direct testing at the concentrations determined by comparison with the methanol standard. The compounds identified by GC–MS and evaluated in this study were 1,8-cineole (Thermo Scientific Chemicals, CAS 470-82-6, Avocado Research Chemicals Ltd. (part of Thermo Fisher Scientific), Lancashire, UK), (±)-α-Terpineol (Ambeed Inc., CAS 98-55-5, Arlington Heights, IL, USA), DL-Menthol (Ambeed Inc., CAS 89-78-1, Arlington Heights, IL, USA), 4-isopropyl-1-methylcyclohexanol (p-Methan-1-ol) (Ambeed, Inc., CAS 21129-27-1, Arlington Heights, IL, USA), and precocene II (AA Blocks, Inc., San Diego, CA, USA). Each of these molecules was prepared as a solution in DMSO solvent.

### 2.3. Cells, Bacteria, and Viruses

In vitro analyses of the impacts of plant extracts on cytotoxicity, inhibition of cancer cell proliferation, and antiviral activity were performed in cultures of HEp-2 [HeLa] cells (ATCC CCL-23), a HeLa-derivative cell line [30]. Mouse coronavirus inhibition experiments were performed in delayed brain tumor-9 (DBT-9) cells, a clonally derived murine astrocytoma cell line generously provided by Mark Denison (Vanderbilt, Nashville, TN, USA). Both DBT-9 and HEp-2 cells were cultured at 37 °C under 5% CO_2_ in complete Dulbecco’s Modified Eagle Medium (DMEM; Corning Company, Corning, NY, USA) supplemented with 10% fetal bovine serum (FBS; Rocky Mountain Biologicals, Crested Butte, CO, USA) and an antibiotic/antimycotic mixture (Corning) consisting of 50 μg/mL penicillin, 50 μg/mL streptomycin, and 2.5 μg/mL amphotericin B. Antibacterial properties were assessed against cultures (maintained strains of Ward’s Science, Rochester, NY, USA) of *Escherichia coli*, *Pseudomonas aeruginosa*, *Staphylococcus aureus*, and *Streptococcus pyogenes* grown in tryptic soy broth (TSB) at 37 °C. Antiviral activity was examined using a recombinant hRSV strain A2, which expresses a far-red fluorescent reporter, monomeric Katushka 2 (mKate2), and murine hepatitis virus (MHV) strain A59 [31]. Strain hRSV A2-mKate was graciously provided by Martin Moore (Emory, Atlanta, GA, USA), and MHV Strain A59 was provided by Mark Denison (Vanderbilt).

### 2.4. Assessment of Cytotoxicity

In preparation of the plates for cytotoxicity analysis, 24 h prior to treatment, approximately 10^4^ HEp-2 or DBT-9 cells were plated per well in a tissue-culture-treated 96-well plate. Cell viability was evaluated in HEp-2 or DBT-9 cells at approximately 70–90% confluency with varying concentrations of either whole-plant extracts or identified plant-derived chemicals for 24 h at 37 °C under 5% CO_2_. Either Water or DMSO was used as a control for comparison since these reagents were used in the preparation of the whole extracts and chemicals, respectively. Untreated wells were used for baseline standardization, and empty wells were used for background subtraction. Additionally, colchicine was used as a positive control for cytotoxicity. After incubation, the supernatant was removed, and culture medium containing an MTS solution (Promega CellTiter 96 Aqueous One Solution Cell Proliferation Assay) was applied and incubated for 2 h. Cell viability was determined by comparing the absorbance of each solution at a wavelength of 490 nm to the corresponding absorbance of a solvent-only control.

### 2.5. Immunofluorescence Microscopy

HEp-2 cells were plated on glass coverslips at a density of 0.1 × 10^6^ cells/well and incubated at 37 °C under 5% CO_2_. After an incubation period of 24 h, the cells were treated with 10% solutions of the plant extract or a pure chemical compound at a concentration consistent with its relative concentration using GC–MS in complete DMEM, incubated for another 24 h, and then fixed, permeabilized, and treated with a FITC-conjugated anti-tubulin antibody (1:200 dilution) (Sigma-Aldrich, St. Louis, MO, USA), TRITC-conjugated Phalloidin (1:300 dilution) (ECM Biosciences, Versailles, KY, USA), and DAPI (4′,6-diamidino-2-phylindole; 1 μg/mL). Cell images were acquired from each treatment condition using a Leica DM5500 fluorescence microscope (Wetzlar, Germany) and assessed for nuclei exhibiting active mitosis and apoptosis, as well as cytoskeletal disruption (on a scale of 1 to 4 by blinded scoring, representing normal cytoskeletal organization to complete cytoskeletal collapse, respectively).

### 2.6. Bacterial Inhibition Assays

Direct plate counts were performed on cultures prepared by inoculating 5 mL of sterile TSB with a starter culture of *E. coli*, *P. aeruginosa*, *S. aureus*, or *S. pyogenes*, along with 5 mg/mL of plant extracts or a water control. The cultures were incubated for 3 h at 37 °C with shaking before serial dilution and plating on TSA for quantification. The percent colony count relative to a water (untreated) control was determined. Antibacterial inhibition in broth culture was assessed by co-incubation of either the whole extract (at concentrations of 0, 0.1, 1, or 5 mg/mL) or pure compounds (varying concentrations) with starter cultures of *S. aureus* or *E. coli* for up to 5 h at 37 °C with shaking. The optical density of the solution was determined by spectrophotometry at a wavelength of 600 nm, and the relative fold change in growth was calculated as a ratio of the absorbance at each time point compared to the initial absorbance of the solution. Starter cultures of *E. coli*, *P. aeruginosa*, *S. aureus*, or *S. pyogenes* were prepared by inoculating 10 mL of sterile tryptic soy broth (TSB) with a single well-isolated colony of the bacteria from a tryptic soy agar (TSA) plate and growing the bacteria at 37 °C with shaking until mid-log growth was observed for each culture.

### 2.7. Viral Inhibition Assays

Infectivity assays were performed by inoculating HEp-2 cells at a confluency of 70% with a mixture containing a multiplicity of infection (MOI) of 0.05 infectious particles per cell of the hRSV strain A2-mKate2 and either the whole extract (at concentrations of up to 2 mg/mL) or pure compounds (at concentrations of up to 100% of levels found in the whole extract). Either water or DMSO was used as a control for comparison, since these reagents were used in the preparation of the whole extracts and chemicals, respectively. The cultures were incubated at 37 °C under 5% CO_2_ for 24 h. After 24 h of incubation, the burden of infection was quantified using a Leica DMIL microscope by counting red fluorescent foci after treatment compared to untreated controls. In addition to the infectivity assays, a replication curve experiment was also performed using hRSV-mKate2 and the whole extracts to identify any changes in virus titer over time by quantifying the virus amount present in collected supernatant aliquots. HEp-2 cells at 70% confluency were infected at an MOI of 0.05 with A2-mKate2 in the presence of 0.2 or 2 mg/mL of plant extracts, and images and supernatants were obtained each day over the course of 3 days. At each time point, an aliquot of the supernatant was frozen at −80 °C. To quantify the amount of virus at each time point, the supernatants were thawed, serially diluted, and applied to HEp-2 cells at 70% confluency in 96-well plates. The number of red fluorescent foci was counted, and the average virus titer was determined.

An inactivation assay for antiviral activity in MHV coronavirus was performed by co-incubating doses of strain A59 at an MOI of 0.01 with varying concentrations (ranging from 0 to 2.5 mg/mL) of peppermint, snakeroot, or a sage (control) extract for 12 h prior to infecting confluent monolayers of DBT-9 cells in 12-well tissue-culture-treated plates by rocking at room temperature for 45 min. After 45 min of infection, the cells were overlaid with a 1:1 mixture of DMEM/2% agarose, and the plates were incubated overnight for plaque formation. Visible plaques were counted, and the percent reductions in virus titer relative to the untreated wells were determined.

### 2.8. Statistical Analyses

RStudio (ver. 3.6.0) with a CAR package was used to perform an analysis of covariance (ANCOVA) for cytotoxicity, antibacterial, and antiviral activities as previously described [29]. Immunofluorescence analyses were analyzed using an ANOVA with significant differences subjected to post hoc multiple comparisons (Tukey’s HSD) to test for differences between specific treatments and a water control. All other statistical analyses, where relevant, used a student’s *t*-test.

## 3. Results

### 3.1. Peppermint (M. × piperita) and White Snakeroot (A. altissima) Extracts Show Limited Effects on the Replication and Proliferation of Cancer Cells In Vitro

Initial experiments were performed to assess whether peppermint and white snakeroot leaf extracts exhibit any cytotoxic properties towards HEp-2 cancer cells in vitro. HEp-2 cells are a HeLa-derived cell line that has been well-described in the literature and is a host cell line for viral infection with hRSV, which will be used later in this study [30]. Cell viability was evaluated using an MTS-based cell viability assay, which quantifies the reduction of the MTS reagent into a colored formazan product driven by metabolically active cells [32]. Aqueous extracts of peppermint and white snakeroot showed no significant signs of cytotoxicity in HEp-2 cancer cells up to concentrations of 25 mg/mL based on an MTS assay (Figure 1a). Cell viability upon treatment with both extracts remained at or above 100% of the cell viability of the water-treated control. Water was selected as a control for this study due to the use of only deionized water in the preparation of the extracts. Interestingly, at higher concentrations, the peppermint extract showed a significantly higher level of cell viability compared to water (*p* < 0.001; ANCOVA). However, after treatment with either peppermint or snakeroot extracts, there appeared to be some visual alterations in cell morphology.

To assess cell morphology and the impacts of treatment, immunofluorescence analysis was performed after treatment with 10 mg/mL solutions of the plant extracts, colchicine (2 μM), or a water control (Figure 1b). The concentration of 10 mg/mL was selected since this concentration exceeds the highest concentrations to be tested in subsequent viral analyses and was expected to reveal any specific cell impacts not detected in the initial cytotoxicity analyses. Colchicine is a well-described inducer of apoptosis, and, consistent with previous studies, there was a notable reduction in extant cells, with significant alterations in cellular structure observed, including rounding and reduced actin and tubulin signal in microscopy images (Figure 1b) [33,34]. In contrast, treatment with either water or a peppermint control did not appear to result in any alterations in growth or cellular morphology. However, cells treated with the snakeroot extract reflected rounding and alterations in cytoskeletal signals that were more similar to colchicine without the corresponding increases in nuclear fragmentation seen in the pro-apoptotic inducer.

Blinded scoring was performed on images of each treatment to quantify any significant differences in cell structure. Colchicine treatment induced a significantly greater percentage (19%) of altered nuclei consistent with apoptosis (*p* = 0.00019) compared to all other treatments (Figure 1c). No significant differences in the percentage of apoptotic nuclei were observed between either peppermint (2%) or white snakeroot (2%) and the water treatment control (1%). Furthermore, no significant differences in mitotic nuclei between all treatments were observed. Cells treated with colchicine (2.33; *p* = 0.0091) and snakeroot (2.17; *p* = 0.02792) exhibited significant differences in the average cytoskeletal disruption compared to the water control (1.17) (Figure 1d). In contrast, no significant difference in the cytoskeleton was observed for peppermint extract treatment (1.33) compared to the water control.

Collectively, these cytotoxicity and immunofluorescence data indicate that the peppermint extract did not result in any significant impacts on cell viability or morphology, while the white snakeroot extract induced significant alterations to cytoskeletal structures without inducing significant apoptosis or cytotoxicity.

### 3.2. Both Peppermint (M. × piperita) and White Snakeroot (A. altissima) Exhibit Significant Antibacterial and Antiviral Activity

Peppermint and white snakeroot extracts were next evaluated for antibacterial activity. An initial experiment was performed to determine if peppermint and white snakeroot extracts inhibit the replication of two species of Gram-negative bacteria, *E. coli* and *P. aeruginosa*, and two species of Gram-positive bacteria, *S. aureus* and *S. pyogenes*. These bacteria were selected to reflect the known structural diversity and metabolic differences among common bacterial pathogens. Direct plate counts were performed on cultures of each bacterial strain after 3 h of incubation at 37 °C in the presence and absence of 5 mg/mL of each aqueous plant extract. Compared to a water control, significant reductions in cultures of *E. coli* (*p* < 0.001) and *S. aureus* (*p* < 0.001) were observed after treatment with the peppermint extract (Figure 2a). Specifically, treatment with peppermint resulted in reductions in *E. coli* and *S. aureus* cultures to 25% (±10%) and 40% (±5%) of the untreated control. However, no significant change in culture was observed for peppermint treatment against either *P. aeruginosa* or *S. pyogenes*. Treatment of the *S. aureus* culture with the snakeroot extract resulted in a significant reduction (*p* = 0.020) to 71% (±8%); however, the snakeroot treatment failed to significantly reduce the cultures of *E. coli*, *P. aeruginosa*, and *S. pyogenes*.

To further characterize the antibacterial activities of peppermint and white snakeroot extracts against *S. aureus* and *E. coli*, bacterial cultures were treated with doses of the plant extracts, and the growth of the bacterial culture was evaluated using optical density over time (Figure 2b). Both peppermint and white snakeroot extracts were capable of significantly reducing the growth of the two bacterial strains at concentrations of 1 and 5 mg/mL (*p* < 0.001). When treated with the lower concentration of 0.1 mg/mL, peppermint (*p* = 0.013) and snakeroot (*p* = 0.006) were capable of significantly reducing *E. coli* growth; however, no significant differences in *S. aureus* growth were observed after treatment with either extract. However, these data together demonstrate that both peppermint and white snakeroot extracts have the capacity to reduce the growth of *E. coli* and *S. aureus* in a dose-dependent manner.

We previously reported that sage (*Salvia lyrata*) extract was a potent inhibitor of hRSV in vitro [29]. The recombinant hRSV strain A2, expressing a red fluorescent reporter, was used to initially investigate whether peppermint and white snakeroot can inactivate the virus. Consistent with previous studies, the sage extract (positive control) significantly reduced the detectable virus infection by 95% (*p* < 0.001) compared to no treatment (Figure 3a). Treatment with peppermint and white snakeroot resulted in significant reductions (*p* < 0.001) of 52% and 43% in the amount of the detectable virus, respectively, compared to an untreated control. From these data, the effective concentrations of sage (0.73 mg/mL), peppermint (2.36 mg/mL), and white snakeroot (2.64 mg/mL) capable of reducing hRSV by 50% (EC50) were calculated.

While these data indicate that hRSV is sensitive to exposure to both peppermint and white snakeroot extracts, it remains unclear how exposure impacts overall viral replication. A subsequent replication curve experiment was performed using an MOI of 0.05, and the amount of released virus into the supernatant was calculated over the course of 3 days of incubation (Figure 3b). At concentrations of both 0.2 and 2 mg/mL for both peppermint and snakeroot treatments, notable reductions in the virus signal were observed as early as 1 day post-infection. The average virus titer by day 3 of infection after treatment with 2 mg/mL of peppermint and snakeroot was 1.4% (*p* = 0.002) and 8.6% (*p* = 0.012) of the untreated control. These data overall demonstrate that peppermint and white snakeroot treatments, at concentrations as low as 0.2 mg/mL, are capable of significantly impairing hRSV replication.

### 3.3. Analysis of the Antiviral Activity of Aqueous Leaf Extracts of Peppermint (M. × piperita) and White Snakeroot (A. altissima) Against a Murine Coronavirus Strain (MHV)

Given the observed antiviral activity of aqueous leaf extracts of peppermint and white snakeroot against hRSV, a follow-up experiment was performed to evaluate if these extracts can inhibit the replication of another virus, murine hepatitis virus (MHV), a mouse coronavirus. MHV is a safe and well-described model for coronavirus replication and is closely related to two common human coronavirus strains, HKU1 and OC43. An initial experiment was performed to first identify any potential cytotoxicity associated with delayed brain tumor-9 (DBT-9) cells, a permissive murine astrocytoma cell line. Compared to the water control, there was no significant reduction in cell viability observed during treatment with either peppermint or snakeroot up to 10 mg/mL (Figure 4a). In contrast, the sage extract control showed significant cytotoxicity beyond the concentration of 2.5 mg/mL (*p* < 0.001). To avoid any concerns over cytotoxicity, a virus inactivation assay was performed with concentrations of up to 2.5 mg/mL (Figure 4b). Consistent with earlier studies reporting anti-coronavirus activity associated with sage extracts, a dose-dependent significant reduction (*p* < 0.001) in the virus was observed, with the EC50 calculated as 0.22 mg/mL. No reduction in titers was observed after treatment with either peppermint or white snakeroot extracts. In fact, at concentrations between 0.5 and 1.25 mg/mL, notable increases in virus titer were observed relative to the water control. These data collectively indicate that peppermint and white snakeroot do not harbor significant antiviral activity against a murine coronavirus.

### 3.4. Chemical Analysis of Peppermint (M. × piperita) and White Snakeroot (A. altissima) Identify Several Major Chemical Constituents

GC–MS analysis was performed to identify the primary chemical constituents present in the peppermint and white snakeroot extracts. The primary hits from the analysis for peppermint were 1,8-cineole, α-Terpineol, menthol, and 4-isopropyl-1-methylcyclohexanol (*p*-Methan-1-ol). For white snakeroot, this was precocene II (Table 1, Figure 5a). These chemicals were subsequently obtained and tested for cytotoxicity and their apparent effects on the replication and proliferation of HEp-2 cancer cells, in the same way as their parent extracts had been tested before. While most of these chemicals had no significant cytotoxic impacts, menthol resulted in significant cytotoxicity (*p* < 0.001) at the approximate concentration present within the peppermint extract, resulting in an 89% decline in percent viability relative to the untreated control (Figure 5b).

When evaluated by immunofluorescence in vitro, cells treated with menthol and cineole displayed reduced cell density and notable changes to cell morphology (Figure 6a). Analysis of nuclear alterations revealed that menthol resulted in a significant increase (*p* = 0.01718) in nuclear alterations consistent with apoptotic activity compared to the DMSO control (Figure 6b). Treatment with either cineole (*p* = 0.0107) or *p*-Methan-1-ol (*p* = 0.00669) triggered significant reductions in the amount of detected mitotic nuclei. No significant nuclear alterations were observed for α-Terpineol or precocene II. Furthermore, blinded scoring of the cytoskeletal structure of the cells after treatment with the chemicals found no significant changes compared to the untreated controls.

The chemicals were next evaluated for antibacterial and antiviral activities. Among the chemicals tested for antibacterial activity, only treatments with cineole or menthol resulted in significant reductions of the replication of *E. coli* and *S. aureus* (Figure 7a). Specifically, cineole treatment at a concentration of 2 mM, consistent with its concentration in the peppermint extract, resulted in a significant reduction in the optical density of both *E. coli* (*p* = 0.009) and *S. aureus* (*p* < 0.001). In addition, cineole resulted in significant reductions in replication at the lower tested concentrations of 0.2 mM (*p* < 0.001) and 0.02 mM (*p* = 0.003) against *S. aureus*. Treatment with menthol at concentrations of 0.5 and 5 mM resulted in significant reductions (all *p* < 0.001) in replication of both *E. coli* and *S. aureus*. No significant changes were observed for any of the tested chemicals against hRSV (Figure 7b).

## 4. Discussion

In this study, we looked at the anticancer, antibacterial, and antiviral properties of peppermint and white snakeroot leaf extracts prepared in aqueous solution. We found that peppermint and white snakeroot extracts exhibit both antibacterial and antiviral activity. Prior to this study, several previous studies reported the ability of peppermint extracts and peppermint oil to inhibit a wide range of bacteria, including *E. coli* and *S. aureus* [12,17]. Consistent with these findings, we also found that the peppermint extract significantly reduces bacterial replication, with activity observed at concentrations as low as 1 mg/mL against *E. coli* and 0.1 mg/mL against *S. aureus*. While these studies were performed using a general-purpose growth media with a normal salt content (0.5% NaCl), it should be noted that differences in salt content or other growth conditions may affect the efficacy of these extracts against different bacterial species. Testing of peppermint and white snakeroot extracts against *P. aeruginosa* and *S. pyogenes* did not result in significant reductions by plate count. These data may suggest that the mechanism of activity associated with declines in *E. coli* and *S. aureus* cultures could be due to a specific molecular interaction rather than a more global chemical effect. Studies investigating mechanisms of resistance have shown that efflux pumps, which are known to be present and vary in nature in both Gram-positive and Gram-negative bacteria, are a common mechanism of chemical resistance to antimicrobials, including many modern antibiotics [35]. In particular, *Pseudomonas aeruginosa*, which has recently been associated with both multi-drug-resistant (MDR) and extensively drug-resistant (XDR) strains, is known to utilize four main sets of efflux pumps associated with antibiotic resistance (MexAB-OprM, MexXY, MexCD-OprJ, and MexEF-OprN) [36]. Likewise, several efflux pumps (such as *Mef*) have been identified in strains of *S. pyogenes*, conferring resistance to antibiotics such as erythromycin and clindamycin [37,38]. The failure of *P. aeruginosa* and *S. pyogenes* to respond to peppermint and snakeroot extracts may be consistent with the ability of strain-specific efflux pumps present in these species to remove cytotoxic compounds present in the extracts.

Despite its therapeutic use for a diverse array of respiratory ailments, very little is known about whether peppermint can inhibit the major pathogen, hRSV. Only one previous study reported supposed activity; however, this study failed to measure hRSV inhibition directly (instead relying on changes in cytopathic effects) and did not provide direct evidence of the findings [16]. In this study, we show that the peppermint extract was able to significantly reduce hRSV infectivity at a dose of 2.5 mg/mL. At this concentration, there were no observed impacts on HEp-2 cell viability or morphology. Several chemicals were identified in the peppermint extract and were subsequently evaluated. As others have reported, pure 1,8-cineole and menthol demonstrated effects on cell viability and antibacterial activity [17]. However, none of the chemicals tested for inhibition of hRSV resulted in significant reductions in viral infectivity. This may suggest either a synergistic effect of some of the chemicals tested or another chemical constituent that was not identified or tested in this study. Future studies may consider the use of replication-based assays, biochemical analyses, and different hRSV strains to better understand the mechanism of action and any specific chemical inhibitions that may exist. Collectively, these findings support continued study of the therapeutic potential of peppermint and its potential to inhibit hRSV and other potential pathogens.

White snakeroot (*A. altissima*) is a native plant found in the Midwestern United States, and nothing has been reported regarding its antimicrobial properties. In this study, we show that the snakeroot extract significantly reduced both *E. coli* and *S. aureus* growth at concentrations as low as 0.1 mg/mL and 1 mg/mL, respectively. Similar to the peppermint extract, the snakeroot extract also significantly reduced hRSV replication in vitro. Analysis of treated HEp-2 cells showed that the snakeroot extract significantly altered the cytoskeleton of cells but failed to result in significant cytotoxicity. These data indicate that the snakeroot extract exhibits significant antibacterial and antiviral properties, which warrant future study. During the chemical analysis of the snakeroot extract, we identified precocene II (also known as ageratochromene II) as a major constituent. To the best of our knowledge, this compound has not been previously associated with snakeroot species, although it has been identified in several other members of the family *Asteraceae*, including *Ageratum conyzoides* and *Calea serrata*, where it appears to disrupt hormonal signaling and may exhibit tissue-specific cytotoxicity [39,40,41,42].

In addition to hRSV, we also tested peppermint and white snakeroot extracts for antiviral activity against MHV, a mouse coronavirus that is a well-described replication model that shares a close genetic relatedness to two human common cold coronaviruses. Previous studies have described peppermint essential oils as harboring significant antiviral activity against SARS-CoV-2 and infectious bronchitis virus (IBV), an avian coronavirus [43,44]. However, it is unclear how the preparation of peppermint affects its activity. We show that aqueous extracts of either peppermint or white snakeroot at concentrations of up to 2.5 mg/mL resulted in no reduction in the virus compared to the untreated control. In fact, there were notable increases in the virus detected at several concentrations. This observation may indicate that the plant extract has positive impacts on cell viability, which promotes more productive virus replication.

Based on the findings of this study, whole peppermint and white snakeroot extracts appear to exhibit broader impacts on both bacterial and viral replication (in the case of hRSV) compared to the individual compounds tested. While individual activities were identified and reported with respect to pure compounds, our findings indicate that there are likely other compounds present or synergistic effects that account for the differences between the whole extracts and pure compounds evaluated. While other solvents and extraction methods have been used in the study of peppermint and white snakeroot, the approach used in preparing the extracts in this study was performed in a traditional manner, using water as a solvent. Future studies may investigate other extraction procedures for antibacterial and antiviral activity. Additionally, the selection of HEp-2 cells as a mammalian cell line for testing was driven largely by their role as a host for hRSV infection. However, the use of just two cancer cell lines represents a key limitation of our conclusions regarding any cytotoxic or morphological impacts associated with extract treatment. The use of other cell lines, including primary non-cancerous cell lines and those derived from other tissues, may provide additional insights into the activity of these extracts. In evaluating the extract impacts on cells, additional approaches and techniques, such as flow cytometry to evaluate nuclear fragmentation, may add clarity to subtle changes that are not detectable through immunofluorescence or MTT-based assays alone. Lastly, the mechanism of action for the activities reported here remains unclear and should be investigated. There remain many exciting and novel avenues with which to build upon and improve the initial findings of this study.

In conclusion, we demonstrate for the first time that aqueous leaf extracts of both peppermint and white snakeroot significantly inhibit hRSV infectivity. Additionally, we show that peppermint and white snakeroot leaf extracts potently restrict the replication of two species of bacteria, *E. coli* and *S. aureus*, while not significantly impacting the replication of *S. pyogenes* and *P. aeruginosa* bacterial cultures. Chemical characterization of these two extracts identified several major chemical constituents. Of these, we show that 1,8-cineole (eucalyptol) and menthol exhibit significant antibacterial activity in peppermint; however, none of the extracts tested resulted in significant activity against hRSV. Prior to this study, neither peppermint nor snakeroot had been shown to exhibit significant activity against mammalian cancer cells. Consistent with these observations, we also did not observe cytotoxicity in either plant extract when tested against two different cancer lines, HEp-2 and DBT-9. Immunofluorescence analysis of the treatment in HEp-2 cells revealed only a significant perturbation in cytoskeletal organization in response to the snakeroot extract. These findings collectively suggest that whole aqueous extracts of peppermint and white snakeroot are not likely to possess antiproliferative activities that warrant further study for anticancer therapies. These studies provide novel insights into the biological activity of aqueous leaf extracts from two common plants found in North America.

## Figures and Tables

**Figure 1 microorganisms-14-00080-f001:**
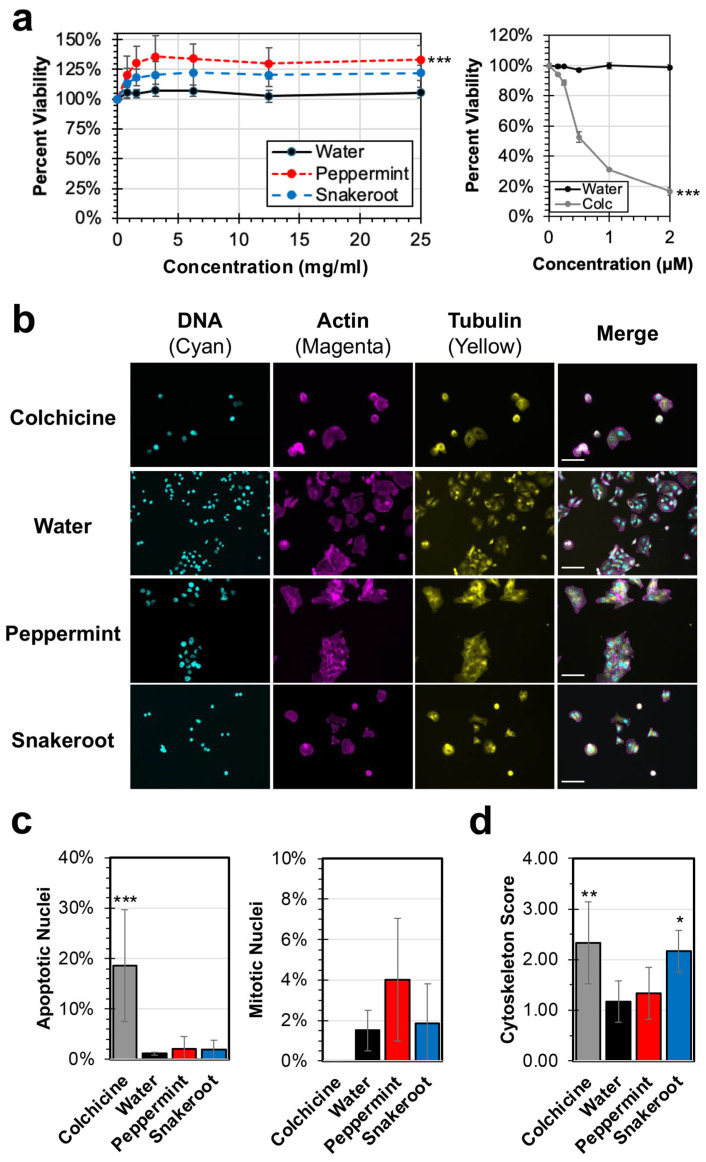
Cell viability and immunofluorescence analysis of HEp-2 cell morphology after treatment with peppermint (*M.* × *piperita*) and white snakeroot (*A. altissima*) extracts. (**a**) HEp-2 cells were incubated with aqueous extracts of peppermint, snakeroot, or a water control (**left**) or colchicine or water (**right**) for 24 h at 37 °C under 5% CO_2_ before cell viability was determined using an MTS assay. The average percent cell viability (±SEM) of 5 experimental replicates is shown. (**b**) HEp-2 cells were incubated for 24 h in complete DMEM containing 10 mg/mL of peppermint extract, snakeroot extract, colchicine (2 μM, a known inducer of apoptosis), or water. After incubation, the cells were fixed, permeabilized, and stained with DAPI (to stain DNA; cyan), TRITC-Phalloidin (actin; magenta), and FITC-conjugated anti-tubulin (tubulin; yellow). Separate and merged image channels are shown for each treatment. Scale bar = 100 μm. (**c**,**d**) Six random images after each treatment were evaluated using blinded scoring to determine the percent of nuclei exhibiting altered nuclear structures (**c**), consistent with either apoptosis (**left**) or mitosis (**right**), or cytoskeletal alteration (**d**), with scores ranging from 0 (normal) to 4 (cytoskeletal destruction). The average percent or cytoskeleton score (±SEM) of the 6 images is shown. An ANCOVA was used to evaluate differences in cytotoxicity and an ANOVA with Tukey’s HSD was conducted for immunofluorescence analyses. Significant differences for all statistical analysis compared to water treatment are reported (*, *p* < 0.05; **, *p* < 0.01; ***, *p* < 0.001).

**Figure 2 microorganisms-14-00080-f002:**
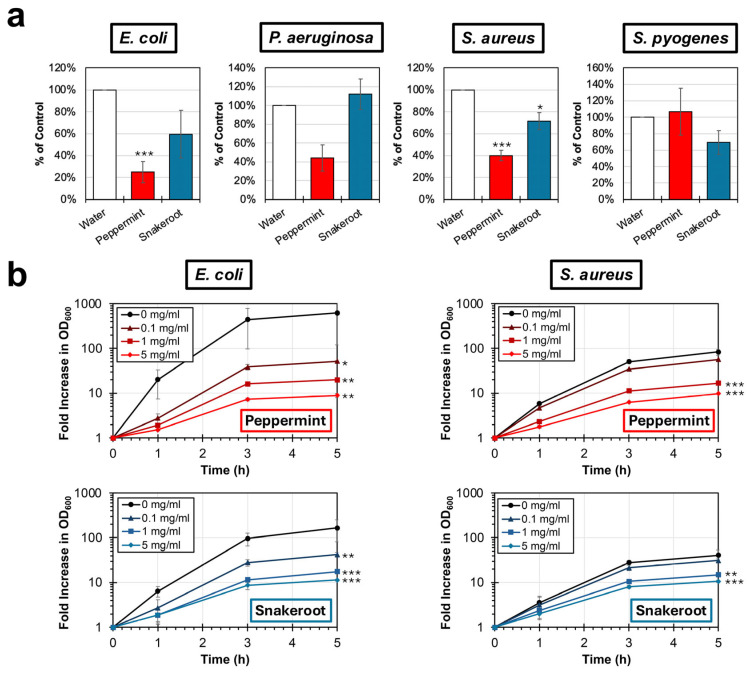
Antibacterial properties of peppermint (*M.* × *piperita*) and white snakeroot (*A. altissima*) extracts. (**a**) Direct plate counts were performed on cultures of *E. coli*, *P. aeruginosa*, *S. aureus*, and *S. pyogenes* after incubation at 37 °C in TSB containing either 0 or 5 mg/mL of extract. The average percent (±SEM; N = 3) of colony count relative to untreated water controls is shown. (**b**) Bacterial cultures of *E. coli* (**left**) and *S. aureus* (**right**) were cultured in TSB at 37 °C in concentrations of peppermint and white snakeroot extracts ranging from 0 to 5 mg/mL. The average fold (±SEM; N = 3) increase in optical density at a wavelength of 600 nm compared to an untreated control is shown. A *t*-test was conducted to evaluate significance in plate counts compared to the water control, and an ANCOVA was used to evaluate significance between treatments and the untreated control for bacterial replication experiments. Significant differences are indicated (*, *p* < 0.05; **, *p* < 0.01; ***, *p* < 0.001).

**Figure 3 microorganisms-14-00080-f003:**
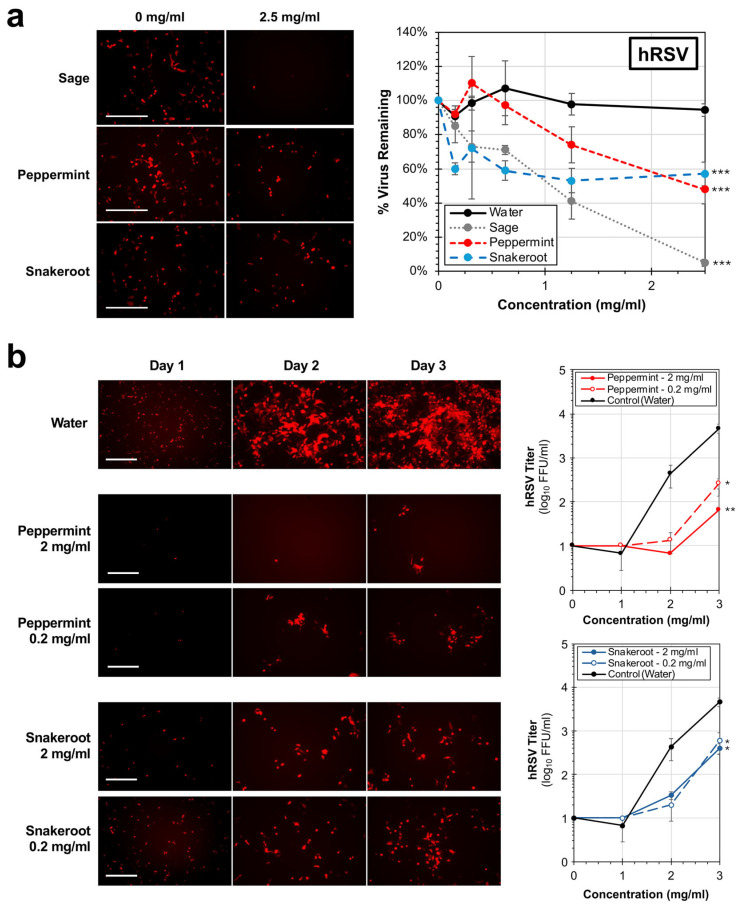
Antiviral properties of peppermint (*M.* × *piperita*) and white snakeroot (*A. altissima*) extracts against human respiratory syncytial virus (hRSV). (**a**) HEp-2 cells were infected at 37 °C with a recombinant hRSV strain A2 expressing a red reporter, mKate2, treated with concentrations of peppermint and white snakeroot extracts ranging from 0 to 2.5 mg/mL. Fluorescent images depicting infected cells (red foci) were obtained after 24 h and are shown on the left (average percent virus remaining (±SEM; N = 3)) compared to an untreated control (shown on the right). Scale bar = 100 μm. (**b**) HEp-2 cells were infected at an MOI of 0.05 with A2-mKate2 in the presence of either water or 0.2 or 2 mg/mL of plant extracts. Images and supernatants were obtained each day. The supernatant was titered on HEP-2 cells, and the average virus titer (right; ±SEM; N = 3) was determined. An ANCOVA was conducted to evaluate significance and significant differences between treatments and the untreated control for all viral experiments. Significant differences are indicated (*, *p* < 0.05; **, *p* < 0.01; ***, *p* < 0.001).

**Figure 4 microorganisms-14-00080-f004:**
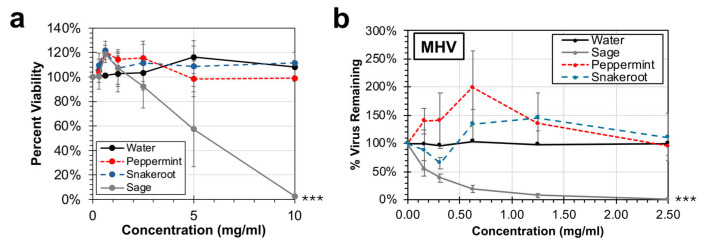
Cell viability of a murine cell line (DBT-9) and susceptibility to inactivation of a murine coronavirus (MHV) after treatment with peppermint (*M.* × *piperita*) and white snakeroot (*A. altissima*) leaf extracts. (**a**) DBT-9 cells were incubated with aqueous extracts of peppermint, white snakeroot, sage extract, or a water control for 24 h at 37 °C under 5% CO_2_. Cell viability was determined using an MTS assay, and the average percent cell viability (±SEM) of 3 experimental replicates is shown. (**b**) DBT-9 cells were incubated for 12 h with MHV strain A59 and varying doses of peppermint, white snakeroot, or sage extract at 37 °C under 5% CO_2_. After exposure, the treated supernatant was titered using a plaque assay on DBT-9 cells to quantify the amount of remaining infectious virus. The average percent of virus remaining (±SEM) compared to the untreated control is shown. An ANCOVA was conducted to evaluate significance for both experiments compared to controls. Significant differences are indicated (***, *p* < 0.001).

**Figure 5 microorganisms-14-00080-f005:**
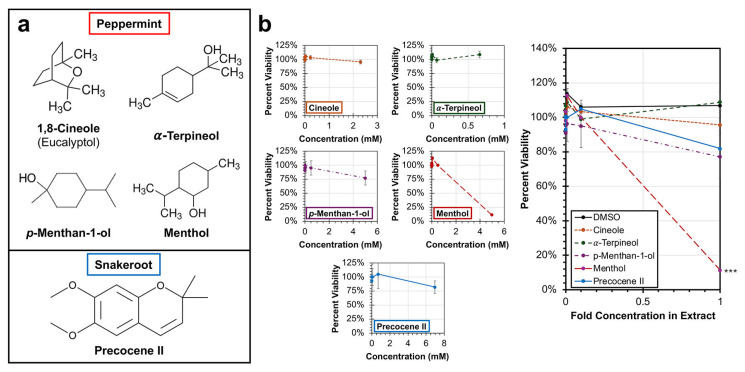
Analysis of cytotoxicity of chemicals identified by GC–MS in peppermint (*M.* × *piperita*) and white snakeroot (*A. altissima*) extracts. (**a**) The structures of the major chemical constituents identified by GC–MS analysis of peppermint and white snakeroot are shown. (**b**) HEp-2 cells were incubated with the individual chemicals or a DMSO control for 24 h at 37 °C under 5% CO_2_ before cell viability was determined using an MTS assay. The average percent cell viability (±SEM) as a measure of the fold concentration relative to the parent extract is shown. An ANCOVA was conducted to evaluate significance, and significant differences are indicated (***, *p* < 0.001).

**Figure 6 microorganisms-14-00080-f006:**
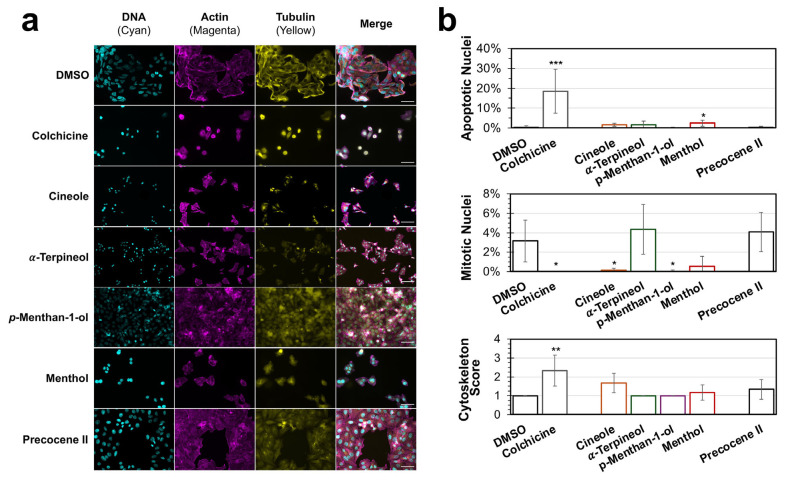
Immunofluorescence analysis of HEp-2 cell morphology after treatment with pure chemical constituents identified in peppermint (*M.* × *piperita*) and white snakeroot (*A. altissima*) extracts. (**a**) HEp-2 cells were incubated for 24 h in complete DMEM containing pure chemicals at the relative concentrations identified by GC–MS, colchicine (2 μM, an inducer of apoptosis), or DMSO. After incubation, the cells were fixed, permeabilized, and stained with DAPI (to stain DNA; cyan), TRITC-Phalloidin (actin; magenta), and FITC-conjugated anti-tubulin (tubulin; yellow). Separate and merged image channels are shown for each treatment. (**b**) Images after each treatment were evaluated by blinded scoring to determine the percent of nuclei exhibiting altered nuclear structure or cytoskeletal alteration, with scores ranging from 0 (normal) to 4 (cytoskeletal destruction). The average percent or cytoskeleton score (±SEM) of the 6 images is shown. Scale bar = 100 μm. A one-way ANOVA with Tukey’s HSD was conducted to evaluate significance, and significant differences for all experiments compared to DMSO treatment are indicated (*, *p* < 0.05; **, *p* < 0.01; ***, *p* < 0.001).

**Figure 7 microorganisms-14-00080-f007:**
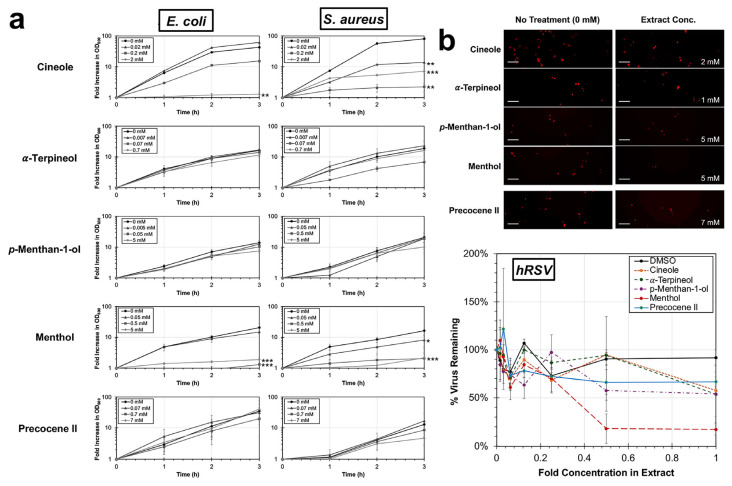
Analysis of antibacterial and antiviral properties of major chemical constituents in peppermint (*M.* × *piperita*) and white snakeroot (*A. altissima*) extracts. (**a**) Bacterial cultures of *E. coli* (**left**) and *S. aureus* (**right**) were cultured in TSB at 37 °C in concentrations approximately 1-fold, 1/10-fold, or 1/100-fold of the amount detected in plant extracts. The average fold (±SEM; N = 3) increase in optical density at a wavelength of 600 nm compared to an untreated control is shown. (**b**) HEp-2 cells were infected at 37 °C with a recombinant hRSV strain, A2, expressing a red reporter, mKate2, treated at concentrations of chemicals ranging from 0 to 1-fold equivalence in the extract. Fluorescent images depicting infected cells (red foci) are shown on the left, and the average percent virus remaining (±SEM; N = 2) compared to an untreated control is shown on the right. Scale bar = 100 μm. An ANCOVA was conducted to evaluate significance and significant differences between treatments and the untreated control for bacterial experiments and between treatment and the water control for viral experiments. Significant differences are indicated (*, *p* < 0.05; **, *p* < 0.01; ***, *p* < 0.001).

**Table 1 microorganisms-14-00080-t001:** Identified major chemical constituents found in peppermint (*M.* × *piperita*) and white snakeroot (*A. altissima*) extracts by GC–MS analysis.

Plant Extract	Chemical ^1^	Approx. Concentration ^2^
Peppermint	1,8-Cineole (eucalyptol)	0.56 M
	α-Terpineol	0.66 mM
	4-isopropyl-1-methylcyclohexanol (*p*-Menthan-1-ol)	5.01 mM
	Menthol	4.97 mM
Snakeroot	Precocene II	6.97 mM

^1^ Identification using the NIST MS library linked to the GC–MS analysis software. ^2^ The approximate concentration of identified chemicals was determined by comparison of the peak of the chemical compared to a spike methanol control of known concentration.

## Data Availability

The original contributions presented in this study are included in the article. Further inquiries can be directed to the corresponding author.

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
