# Peer review of "Aqueous Leaf Extracts of Peppermint (Mentha × piperita) and White Snakeroot (Ageratina altissima) Exhibit Antibacterial and Antiviral Activity"

_microorganisms, 2025, doi:10.3390/microorganisms14010080_

Round 1
Reviewer 1 Report (Previous Reviewer 1)
Comments and Suggestions for Authors
The authors resolved my requests within the allotted time, so the manuscript could be published.
Comments on the Quality of English LanguageI'm sorry, but I'm not qualified to comment on the quality of the English language.
Author Response
We thank the reviewer for their time and consideration.
Reviewer 2 Report (Previous Reviewer 2)
Comments and Suggestions for Authors
- Overall, many spelling and typos which need to be looked through and carefully corrected throughout the article. E.g., Line 210: 5mg/mL; Line 254: ANOVA, and so on. This includes not just the main text, but also the figure labels and figure legends.
- Please supplement the source of origin for the cells, viruses and bacterial strains used in the methodology section.
- Please supplement details of source of antibodies (company, catalog code etc.) used for the immunofluorescence assays in the methods.
- Cytotoxicity assay in the methods section is not clearly written, lacking in details such as the plates used to seed the cells, cell count, controls used etc.
- Bacterial and Viral inhibition assays in the methods section is unclear and lacks details. Please rewrite to improve clarity.
- How did the authors determine to use only the concentration of 10mg/mL of the plant extracts for the cell morphology imaging? What is the concentration of Colchicine used in the assay? Also 10mg/mL? Where is the positive control for Figure 1a? Please supplement it.
- Figure legend line 296, please check and correct the errors. (d) seems to be missing.
- What is the rationale for the choice of bacterial lines in this study? Were the conditions tested in high and low salt?
- It would be good to redo Figure 2a to make it clearer to the readers what you are trying to show. Please supplement the Positive controls for the experiments in Figure 2.
- Please clarify Figure 3b, what is the control? Please supplement images and data from both positive and negative controls.
- Did the authors perform the experiment of the cytotoxicity, antibacterial, and antiviral assays of their identified chemicals from the plant extracts with the whole extract for comparison? If yes, please supplement the data. Please also include the positive and negative controls.
- Not sure why is Figure 7 presented at the end, the article needs to be rearranged to improve its cohesiveness. Please supplement controls for Figure 7.
- Were the individual identified compounds from the whole extract more effective in the antibacterial and antiviral properties compared to the whole extract itself? Were there synergistic effects of the individual compounds? What are their mechanisms of actions giving rise to these antibacterial and antiviral effects?
Author Response
We want to thank the reviewer for their careful reading and helpful suggestions for improvements to the paper. We have addressed all of the comments and made several significant revisions in response. See responses in bold below.
Comments and Responses
Overall, many spelling and typos which need to be looked through and carefully corrected throughout the article. E.g., Line 210: 5mg/mL; Line 254: ANOVA, and so on. This includes not just the main text, but also the figure labels and figure legends.
A thorough check for spelling, grammar, and content was performed throughout.
Please supplement the source of origin for the cells, viruses and bacterial strains used in the methodology section.
The source of origin for all cells, viruses, and bacteria have now been provided along with companies and addresses, where applicable (see lines 152 – 167).
Please supplement details of source of antibodies (company, catalog code etc.) used for the immunofluorescence assays in the methods.
The source (and company information/address) of the anti-tubulin antibody and phalloidin used for the immunofluorescence assays has been provided (see lines 184 – 190).
Cytotoxicity assay in the methods section is not clearly written, lacking in details such as the plates used to seed the cells, cell count, controls used etc.
We have largely rewritten the methods section detailing the cytotoxicity assay and have now included specific information concerning cell seeding, plates used, and controls (see lines 169 – 181).
Bacterial and Viral inhibition assays in the methods section is unclear and lacks details. Please rewrite to improve clarity.
We have rewritten both the bacterial and viral inhibition assay methods for clarity and depth. These sections now include more detailed information on starter culture preparation (bacteria), bacterial culture conditions, and controls (for both bacterial and viral experiments). See lines 195 – 237.
How did the authors determine to use only the concentration of 10mg/mL of the plant extracts for the cell morphology imaging? What is the concentration of Colchicine used in the assay? Also 10mg/mL? Where is the positive control for Figure 1a? Please supplement it.
We have included a statement in the Results section which explains our rationale for use of the concentration of 10 mg/ml for the immunofluorescence experiments (see lines 266 – 269). We have also included the concentration of Colchicine used (2 mM) in both the Results (see lines 264 – 266) and figure legends (Figs. 1 and 6). Figure 1a now includes a new panel reflecting the cytotoxicity of colchicine compared to a water control.
Figure legend line 296, please check and correct the errors. (d) seems to be missing.
We apologize for this mislabeling and have now corrected it. See Figure 1 legend.
What is the rationale for the choice of bacterial lines in this study? Were the conditions tested in high and low salt?
We have now included a statement in the Results explaining a methodology for selecting bacterial strains evaluated in this study (see lines 321 – 322). The bacteria were evaluated strictly in low salt conditions and we have made a note of this in the Discussion (see lines 513 – 516) and have also included a statement about the potential impacts of salt on the findings of this study and the biological potential of the extracts.
It would be good to redo Figure 2a to make it clearer to the readers what you are trying to show. Please supplement the Positive controls for the experiments in Figure 2.
We have remade Figure 2a as a series of four separate panels for the four strains of bacteria tested and have included the control on each panel.
Please clarify Figure 3b, what is the control? Please supplement images and data from both positive and negative controls.
We have revised Figure 3b panels as well as the figure legend with the identify of the control used in this experiment.
Did the authors perform the experiment of the cytotoxicity, antibacterial, and antiviral assays of their identified chemicals from the plant extracts with the whole extract for comparison? If yes, please supplement the data. Please also include the positive and negative controls.
The whole plant extracts were not directly tested side-by-side with the chemicals within the same experiments. We recognize the limitations in making comparisons between experiments and have addressed this in several places in our discussion.
Not sure why is Figure 7 presented at the end, the article needs to be rearranged to improve its cohesiveness. Please supplement controls for Figure 7.
We have moved Figure 7 and the corresponding Results section to now Figure 4 (before the pure chemical analysis). All subsequent Figures have been shifted to reflect this change. Additionally, controls have been added to the Figure 7 (see 7b). A side note, we also considered moving these experiments upon our previous submission and agree that moving the MHV studies earlier does help address some of the cohesiveness concerns.
Were the individual identified compounds from the whole extract more effective in the antibacterial and antiviral properties compared to the whole extract itself? Were there synergistic effects of the individual compounds? What are their mechanisms of actions giving rise to these antibacterial and antiviral effects?
We have expanded the discussion to address these questions directly and pose relevant future directions to this work (see lines 578 – 598).
Round 2
Reviewer 2 Report (Previous Reviewer 2)
Comments and Suggestions for Authors
Authors have addressed the major concerns.
This manuscript is a resubmission of an earlier submission. The following is a list of the peer review reports and author responses from that submission.
Round 1
Reviewer 1 Report
Comments and Suggestions for Authors
The manuscript was completed and improved according to the recommendations.
Comments on the Quality of English LanguageI am not qualified in this regard.
Author Response
We thank the reviewer for their time and consideration of this manuscript.
Reviewer 2 Report
Comments and Suggestions for Authors
This is a resubmission article. From the previous reviews and comments, substantial changes have been made to improve on the manuscript. Some suggestions for the revised manuscript:
-The abstract should also be edited to clearly reflect the scope of this study. The aims is not clear and the conclusions are also not highlighted well.
-It will be good to clearly indicate what solution were the plant samples dissolved in for each study. Is only deionized water being used?
Author Response
Comment: The abstract should also be edited to clearly reflect the scope of this study. The aims is not clear and the conclusions are also not highlighted well.
Response: We have amended the abstract to more clearly reflect the scope, aims, and conclusions of this study.
Comment: It will be good to clearly indicate what solution were the plant samples dissolved in for each study. Is only deionized water being used?
Response: Deionized water was the only solvent used for plant sample dissolution. We have included a statement in the introduction and the results to clarify this point.
We thank the reviewer for their helpful feedback and time and consideration of this manuscript.